# Transmembrane Amino Acid Transporters in Shaping the Metabolic Profile of Breast Cancer Cell Lines: The Focus on Molecular Biological Subtype

**DOI:** 10.3390/cimb47010004

**Published:** 2024-12-25

**Authors:** Elena I. Dyachenko, Lyudmila V. Bel’skaya

**Affiliations:** Biochemistry Research Laboratory, Omsk State Pedagogical University, 644099 Omsk, Russia; dyachenko.ea@gkpc.buzoo.ru

**Keywords:** transmembrane amino acid transporters, amino acids, cell lines, breast cancer, molecular biological subtype, metabolism

## Abstract

Amino acid metabolism in breast cancer cells is unique for each molecular biological subtype of breast cancer. In this review, the features of breast cancer cell metabolism are considered in terms of changes in the amino acid composition due to the activity of transmembrane amino acid transporters. In addition to the main signaling pathway PI3K/Akt/mTOR, the activity of the oncogene c-Myc, HIF, p53, GATA2, NF-kB and MAT2A have a direct effect on the amino acid metabolism of cancer cells, their growth and proliferation, as well as the maintenance of homeostatic equilibrium. A distinctive feature of luminal subtypes of breast cancer from TNBC is the ability to perform gluconeogenesis. Breast cancers with a positive expression of the HER2 receptor, in contrast to TNBC and luminal A subtype, have a distinctive active synthesis and consumption of fatty acids. It is interesting to note that amino acid transporters exhibit their activity depending on the pH level inside the cell. In the most aggressive forms of breast cancer or with the gradual progression of the disease, pH will also change, which will directly affect the metabolism of amino acids. Using the cell lines presented in this review, we can trace the characteristic features inherent in each of the molecular biological subtypes of breast cancer and develop the most optimal therapeutic targets.

## 1. Introduction

Dysregulation of cellular metabolism is a common feature of cancer [1]. First, this concerns the disruption of the regulation of glucose and amino acid utilization. As substrates for protein synthesis, amino acids are important sources of energy and nutrition, second only to glucose [2]. A rich supply of amino acids allows the tumor to maintain its proliferative capacity by participating in energy production, nucleoside synthesis, and maintaining the oxidation–reduction balance in the cell [3]. Studying metabolic pathways in cancer cells is a key aspect in developing new approaches for diagnosis, treatment, and prevention of oncological diseases. It enables the identification of vulnerable points and targets for identifying new drugs and therapeutic strategies [4]. Some of the current research includes searching for molecular markers, developing inhibitors of key enzymes, and using new technologies for visualization and analysis of metabolic processes in tumors [5].

Breast cancer is the most frequently diagnosed cancer among women [6]. Breast cancer is a heterogeneous disease, with genetic subtypes of breast cancer defined based on the coding genome [7]. There are five main molecular biological subtypes of breast cancer. The metabolism of amino acids in breast cancer cells is unique to each molecular biological subtype of breast cancer. This review considers the features of cell metabolism in a particular subtype of breast cancer in terms of changes in the amino acid composition due to the activity of transmembrane amino acid transporters.

The main research tool in studying the features of molecular biological subtypes of breast cancer is cell lines. Approximately 80 breast cancer cell lines have been developed and tested, each corresponding to a certain molecular biological subtype of breast cancer [8]. Examples of all known cell lines corresponding to each molecular subtype of breast cancer are provided. Even though this review mainly focuses on amino acid metabolism, it is also worthwhile to briefly discuss the features of biochemical processes in breast cancer cells. This approach reveals a complex yet structured system linking the activities of certain genes, the hormonal system, and individual signaling pathways that are inextricably linked with amino acid metabolism.

The aim of the work was to create a holistic and structured understanding of the features of metabolic processes occurring in different subtypes of breast cancer, tracing the flow of the main sources of building material and energy in the form of amino acids through the activity of transmembrane amino acid transporters.

## 2. Main Metabolic Features of Molecular Biological Subtypes of Breast Cancer

In this paper, we grouped all breast cancer subtypes into three subgroups: luminal subtypes, HER2-positive breast cancer, and triple-negative breast cancer (TNBC). Luminal subtypes of breast cancer include luminal A, luminal B (HER2-), and luminal B (HER2+). Luminal A and B (HER2-) subtypes have a positive expression of estrogen and progesterone receptors and a negative expression of HER2. Luminal B (HER2-) subtype, in contrast to the luminal A subtype of breast cancer, has a high index of proliferative activity Ki-67. The luminal B (HER2+) subtype of breast cancer is the most complex among all luminal subtypes as it is characterized by the positive expression of hormone receptors and HER2. Despite the uniqueness of the characteristics of the luminal B (HER2+) subtype of breast cancer and its isolation from other luminal subtypes, we consider all hormone-positive subtypes together. Magometschnigg et al. showed that free estrogen affects insulin signaling, thereby activating the growth and proliferation of cancer cells in luminal subtypes of breast cancer [9]. Estrogens, namely estradiol, are the most powerful stimulator of estrogen receptors (ER) in hormone-positive breast cancer and are also capable of inducing the expression of insulin receptors. In this case, there is an easier entry of glucose into cells and a decrease in the activity of adipose tissue lipase [10]. At high glucose levels, in addition to insulin receptors, insulin-like growth factor 1 (IGF-1) is activated (Figure 1). This factor binds to its complementary receptor IGF-1R, which activates two signaling pathways: phosphoinositide 3 kinase/Akt/mammalian target of rapamycin (PI3K/Akt/mTOR) and mitogen-activated protein kinase/extracellular signal-regulated kinase (MAPK/ERK). The signaling of these pathways triggers the growth and proliferative activity of cancer cells [11]. The main component among the listed signaling cascades is mTOR. It is responsible for cellular metabolism, the activity of which is also controlled by the level of extracellular and intracellular amino acids, affects cell growth, apoptosis, and autophagy [12]. In addition, mTOR activation affects the functional state of 40S ribosomal S6 kinase 1 (S6K1), which is involved in phosphorylation (ER-α) on serine 167. This leads to activation of genes at the transcriptional level that are sensitive to estrogen [13]. Thus, estrogen-signaling targets in luminal subtypes of breast cancer include ER itself, IGF1, and its receptor IGF-1R and mTOR.

A striking difference between the luminal A subtype of breast cancer and TNBC is the presence of active gluconeogenesis due to the high expression of fructose-1,6-bisphosphate. Fructose-1,6-bisphosphate is the limiting and key enzyme in gluconeogenesis [14].

The molecular biological subtype of breast cancer with positive HER2 expression has high proliferative activity. Increased HER2 expression and phosphorylation of this receptor activates the PI3K/AKT signaling pathway. Activation of PI3K/AKT increases GLUT4 expression, which promotes glucose penetration into the cell and its further glycolysis. It was noted that active glucose transport through GLUT4 and increased glycolysis are observed at the early stages of breast cancer with positive HER2 expression [15,16]. HER2 expression also affects the activity of 6-phosphofructo-2-kinase/fructose-2,6-bisphosphatase 3 (PFKFBP3). This leads to increased anaerobic glycolysis and resistance to trastuzumab [17]. During active glycolysis, pyruvate is actively converted into lactate by lactate dehydrogenase-A (LDHA). Glycolysis occurs via the anaerobic pathway. Accumulated lactate helps maintain the anaerobic glycolytic process and tumor cell growth [18,19,20,21].

Positive HER2 expression in breast cancer cells demonstrates an increased level of glutamine and fatty acid consumption [22,23]. At the transcriptional level, there is an increase in the expression of lipogenic enzymes [24]. The accumulation of 2-hydroxyglutarate leads to DNA hypermethylation [25,26], which is a subtype-specific process [27]. A recent study reported that DNA hypermethylation in TNBC reflects a suppressed DNA state and is not associated with tumor progression. However, DNA hypermethylation in the luminal subtype affects gene expression; therefore, it can contribute to tumor progression [28]. A distinctive feature between TNBC and HER2-positive breast cancer is the expression of glucose transporters. TNBC, as the most invasive type of cancer, has the highest level of glucose transporter-1 (GLUT-1) expression compared to other subtypes [29,30]. As noted earlier, with positive HER2 expression, GLUT-4 has the highest activity.

TNBC cells exhibit high dependence on glycolysis, altered glucose, fatty acid, and amino acid metabolism, which contributes to increased cellular bioenergetics demands as the cancer continues to proliferate and metastasize [31]. Interestingly, epidermal growth factor receptor tyrosine kinase receptor (EGFR) and mesenchymal–epithelial transition (MET) signaling are closely associated with metabolic changes in TNBC cells [32]. The major signaling pathway dependent on and regulating cellular metabolism and amino acid flux is PI3K/Akt/mTOR. Oncogene c-Myc, hypoxia-inducible factor 1-alpha (HIF-1α), p53, erythroid transcription factor (GATA2), nuclear factor kappa-light-chain-enhancer of activated B cells (NF-kB), and Methionine Adenosyltransferase 2A (MAT2A) are also involved in amino acid metabolism, growth, proliferation, and homeostatic balance maintenance in cancer cells. Transcription factors such as HIF, c-Myc, and p53 are able to modulate the expression and activity of glucose transporters and enzymes involved in glycolysis and the pentose phosphate pathways (PPP) or the tricarboxylic acid cycle (TCA) [33].

## 3. Luminal Subtypes of Breast Cancer

The main amino acid transporters in luminal subtypes of breast cancer are shown in Figure 2.

As can be seen from Figure 2, only three families of amino acid transporters are well studied and mainly encountered. The first family is known as the family of high-affinity glutamate and neutral amino acid transporters. An increased expression of transmembrane amino acid transporters was found among solute carrier family (SLC) SLC1A1, SLC1A2, SLC1A3, SLC1A4, SLC1A6, and SLC1A7. All of the listed transporters mainly transport negatively charged amino acids, with the exception of SLC1A4. SLC1A4 transports small neutral amino acids. Thus, in the luminal A subtype of breast cancer, the level of cysteine and its oxidized form, cystine, is increased. Cystine is the product of methionine conversion through the formation of an intermediate product, homocysteine. High levels of cystine, with a deficiency of B vitamins in the blood plasma, act as an inhibitor of the methyltransferase enzyme. As a result, the process of DNA methylation and regulation of the expression of genes responsible for oncogenesis becomes ineffective [34]. High cysteine levels affect the activity of cysteine proteases cathepsin B and L. In addition to high cysteine levels, the decreased expression and activity of endogenous inhibitors of cathepsin B and L have been demonstrated in various types of cancer, including luminal A subtype breast cancer [35]. Tumor cells require aspartate for proliferation [36]. Under normal conditions, a pathologically unchanged cell synthesizes aspartate by oxidative phosphorylation (OXPHOS) [37]. In turn, cancer cells face metabolic limitations in aspartate synthesis via OXPHOS. In this case, tumor cells obtain aspartate through increased activation of expression of aspartate transporters SLC1A1, SLC1A2, SLC1A3, SLC1A6, and SLC1A7 [38,39] or its conversion from asparagine [40]. Glutamate transport is carried out in this case together with aspartate via the above-mentioned transmembrane amino acid transporters. Glutamate, along with aspartate, participates in anabolic processes, promotes DNA synthesis, and lipid and protein biosynthesis. Aspartate and glutamate participate in anabolic processes. This statement is consistent with previous reports showing that the bulk of the cell’s carbon mass is formed not by glucose and glutamine but by amino acids [41]. The SLC1A4 transporter in luminal breast cancer carries small neutral amino acids (alanine, serine, cysteine, threonine), but mainly transports serine and glycine [42]. One study showed that a high percentage of mutp53 missense mutations is observed in breast cancer. It was shown that mutp53 correlates with active glucose consumption [43]. Under glucose deficiency conditions, serine/glycine is actively loaded into the cell via the overexpressed SLC1A4 transporter to initiate gluconeogenesis. Due to the high content of intracellular serine and glycine, cancer cells maintain the redox balance and proliferation of cancer cells [44]. To activate mTORC1 and PI3K/Akt signaling, sufficient amino acids, especially leucine, arginine, and glutamine, are required [45,46,47].

In the luminal B subtype, a decrease in the level of alanine and leucine was shown regardless of HER2 expression. Alanine forms a favorable tumor microenvironment in metastatic breast cancer [48]. Alanine decrease occurs due to the inhibition of the enzyme GPT2 (alanine transaminotransferase-2), which catalyzes the reaction of converting glutamate and pyruvate to alanine [49]. Lower leucine levels may be due to high expression of leucine aminopeptidase 3 (LAP3) in breast cancer tissues. LAP3 is an exopeptidase that catalyzes the hydrolysis of leucine residues at the amino terminus of a protein or peptide substrate [50,51]. LAP3 is also involved in the proliferation, migration, invasion, and angiogenesis of breast tumor cells. It enhances the motility and invasion of breast cancer cells by activating multiple signaling pathways. Lieu et al. showed that the amino acid profile is affected not only by the presence of breast cancer, but also by risk factors for its development. Thus, a large number of births correlated with high levels of glutamate and histidine and breast cancer; with age, the content of ornithine increased in patients with breast cancer. Earlier menarche and high levels of glutamic acid were significantly higher in patients with the luminal B subtype of breast cancer [52].

Higher glycine concentrations together with lactate were correlated with poor prognosis and low survival rates among ER-positive breast cancer patients [53,54,55]. Cao et al. showed the opposite effect, whereby suppression of glycine content resulted in decreased proliferative activity [56]. This example once again demonstrates the dependence of mTOR signaling and cell proliferative activity on amino acid sufficiency [57].

A distinctive feature of the luminal B subtype of breast cancer is the active consumption of fatty acids instead of amino acids. In this subtype of breast cancer, the main source of energy production is fatty acid metabolism and lipogenesis [58]. Such tumors are characterized by an increased content of de novo fatty acid synthesis products, such as palmitate-containing PtdCho. Such cells acquire a lipogenic phenotype [59].

Of the sodium- and chloride-dependent neurotransmitter transporter family, two transporters have increased expression in luminal subtypes of breast cancer. The transmembrane transporter SLC6A14 has the highest expression in luminal subtypes of breast cancer. A link was established between SLC6A14 expression and positive ER expression both in primary breast cancer cells and in the MCF cell line [60]. The SLC6A14 amino acid transporter is known for transporting all neutral and cationic amino acids. Highly concentrated amino acid transport consists of a transmembrane gradient formed due to the concentration of Na^+^ and Cl^−^. It is noteworthy that SLC6A14 can unidirectionally transport amino acids into the cell without exchanging them for other amino acids [61]. This mechanism provides cells with a constant and independent influx of amino acids, which are required in large quantities by cancer cells during growth and proliferation. Of the many amino acids transported by the transmembrane carrier SLC6A14, glutamine, leucine, and arginine should be singled out. Glutamine is an amino acid that participates in the biosynthesis of nucleotides and is a substrate for glutaminolysis. Glutamine is converted into glutamate by the enzyme glutaminase. Glutamate, in turn, is converted into alpha-ketoglutarate (a-KG) and then into oxaloacetate. Oxaloacetate with pyruvate form phosphoenolpyruvate, which is the fourth stage of gluconeogenesis [62]. Leucine is one of the main activators of mTOR, which triggers cell proliferation. Arginine is an essential amino acid for cancer cells [63]. SLC6A14 activity depends on c-Myc expression through miR-23a inhibition [64]. The increased activity of miR-23a and miR-23b-3p suppresses the expression of the SLC6A14 transporter protein [65]. The increased activity of heat shock proteins has been found in many types of cancer, including breast cancer. They participate in the correct folding of proteins, including the SLC6A14 transporter protein [66]. In one study, the suppression of activity and deletion of SLC6A14 were observed, which led to a decrease in the flow of essential amino acids into the cancer cell. This led to the inhibition of mTOR and the weakening of HIF-1α phosphorylation [60]. The functioning of SLC6A14 and SLC1A2 transporters is interconnected. With the increased expression of miR-23b-3p, inhibition of SLC6A14 activity occurs, but the activity of SLC1A2 significantly increases. As is known, SLC1A2 provides an influx of aspartate and glutamate, which also supports the proliferation of cancer cells. Such data were obtained when studying the metabolism of cancer cells in breast cancer patients with resistance to endocrine therapy [67]. It is worth noting that the transcription factor GATA2, which is also responsible for the progression of breast cancer, increases the expression of miR-23b-3p [68]. SLC6A15 has also been shown to have increased activity in luminal subtypes of breast cancer, although research in this area is lacking. The SLC6A15 transporter protein shows the greatest preference for transporting branched-chain amino acids (valine, leucine, isoleucine), as well as methionine [69].

Among the cationic amino acid transporter/glycoprotein-associated family, the overexpression of SLC7A1, SLC7A2, SLC7A8, SLC7A9, SLC7A11 has been described [70,71,72]. The increased expression of SLC7A5 has been described in all molecular biological subtypes of breast cancer [73]. Thus, SLC7A1 and SLC7A2 are cationic carriers of the amino acids arginine, histidine, and lysine. Arginine is an essential amino acid for tumor cells. It is involved in the formation of NO from NOS, thereby stimulating the growth of tumor cells and their ability to survive [74]. The increased expression of SLC7A1 and SLC7A2 is associated with a poor prognosis of breast cancer [75,76,77].

## 4. Positive HER2 Expression in Breast Cancer

Among the cationic amino acid transporter/glycoprotein-associated family, SLC7A5, SLC7A7, and SLC7A8 are highly expressed in luminal subtypes of breast cancer, as well as SLC7A9 and SLC7A10, the activity of which is observed exclusively in the case of positive HER2 expression. The expression of SLC1A5 and SLC7A5 is coupled with, but not required for, the function of individual transporter proteins [78] (Figure 3). The SLC7A10 transporter is highly expressed in adipocytes. It mainly transports alanine, serine, and cysteine. One study showed that the reduced expression of SLC7A10 leads to the accumulation of reactive oxygen species due to the lack of amino acids involved in antioxidant protection. In an experiment on mice, it was shown that blocking SLC7A10 led to an increase in fat mass [79].

It is suggested that the transporters of the neutral amino acid carrier family, sodium-coupled transporters in system A and system N, SLC38A2 and SLC38A5, are overactive in HER2-positive luminal B subtype breast cancer. This suggestion is based on the active uptake of glutamine by cancer cells through these transporters [80]. More studies are needed on the role of neutral amino acid carrier family, system A and system N, in breast cancer.

## 5. Triple-Negative Breast Cancer (TNBC)

The main amino acid transporters in TNBC are shown in Figure 4.

The SLC3A1 transporter belongs to the family neutral and basic amino acid transport protein (rBAT). The relationship between SLC3A1 and TNBC was confirmed using the MDA-MB-231 cell line, which corresponds to the basal type B-like TNBC. It is known that the activity of SLC 3A1 and SLC7A11 is coupled in the transport of cysteine. This transporter belongs to the sodium-independent transporters of cysteine, neutral, and dibasic amino acids, which contribute to oncogenesis. As mentioned earlier, cysteine increases glutathione synthesis, thereby reducing reactive oxygen species (ROS) levels and oxidative stress inside the tumor cell. SLC3A1 also signals the activation of AKT [81].

Among the sodium- and chloride-dependent neurotransmitter transporters, SLC6A4 and SLC6A9 have been shown to be overexpressed in TNBC. The transmembrane transporter SLC6A4 is known to be involved in the transport of serotonin and glutamine. Cancer cells, in this case, require sufficient intracellular glutamine to maintain the redox balance by synthesizing glutathione, stimulating the TCA through glutamate formation via GSL1 (glutaminase 1), and acts as a nitrogenous substrate for the synthesis of new amino acids [82,83]. Notably, c-Myc can bind to the miR-23a and miR-23b promoters and suppress their activity, thereby stimulating the activity of GLS1 and SLC6A4 [64]. The SLC6A9 transporter ensures the entry of glycine into the cell and prevents it from hyperosmolarity [84]. In one study, a profile of the metabolic consumption and release of individual cancer cell lines was compiled [85]. It was shown that glycine is actively synthesized de novo and taken up by highly proliferating cells but not by cells with no proliferative activity. It was also shown that increased expression of genes encoding mitochondrial enzymes required for glycine synthesis correlated with a worse prognosis for survival among patients with breast cancer [86]. Together with glycine, SLC6A9 transports serine into the cell. Serine stimulates the growth of cancer cells by replenishing the carbon pool [87]. It has been shown that the activity of enzymes such as phosphoserine aminotransferase 1 (PSAT1), phosphoglycerate dehydrogenase (PHGDH), and phosphoserine phosphatase, which catalyze the synthesis of serine de novo, have increased expression in breast cancer [88,89].

In TNBC, the increased activity of transmembrane transporters SLC7A1, SLC7A5, and SLC7A11 of the cationic amino acid transporter/glycoprotein-associated family was noted. The activity of SLC 1A5 and SLC7A11 is coupled. The antibody 7A11 is an antiporter. Through it, cystine enters the cell. As a result of biochemical reactions, cysteine is converted into cystine. Then, through the 1A5 transporter, cystine is exchanged for glutamate. As noted above, SLC7A5 expression is observed in all molecular biological subtypes of breast cancer. As was mentioned before, in the HER2+ subtype of BC, the activity of SLC7A5 is coupled with SLC1A5. The deep mechanism of this coupled faction still has not been investigated. In TNBC, the SLC7A5 transporter is the only studied and experimentally confirmed source of methionine. The amino acid methionine is involved in the methylation of histones and DNA. Hypermethylation due to methionine occurs as follows. Upon activation of p65, the NF-kB complex is activated and moved into the cell nucleus and hyperactivates MAT2A at the transcriptional level [90]. This enzyme catalyzes the formation of SAM from methionine [91,92]. Further, enzymes such as lysine methyltransferase (LMT) and DNA methyltransferase (DNMT) use SAM for methylation of DNA and histones. Due to this, some tumor suppressor proteins are suppressed. These include the gene of estrogen receptor 1 (ER1), breast cancer gene 1 (BRCA1), tissue inhibitor of metalloproteinase-1 (TIMP1), and phosphatase and tensin homolog deleted on chromosome 10 (PTEN) [93]. Thus, PTEN, which inhibits the activity of PI3K/Akt, is hyper methylated by the enzyme DNA methyltransferase (DNMT), PI3K/Akt becomes active, which leads to the growth and proliferation of cancer cells, as well as the suppression of apoptosis [94,95]. In addition, PI3K/Akt suppresses miR-14b activity, which is responsible for preventing NF-kB translocation to the cell nucleus and triggering histone and DNA hypermethylation [96,97]. Another study showed that high methionine levels stimulate metastasis in TNBC [98].

The monocarboxylic acid transporter family includes SLC16A1, SLC16A3, SLC16A7, SLC16A8, and SLC16A10, which are mainly involved in the transport of lactate, pyruvate, aromatic amino acids (phenylalanine, tryptophan, and tyrosine), and ketone bodies. It is known that the monocarboxylate transporter, along with the glucose transporter GLUT1, has increased expression in TNBC, since there is excessive absorption of glucose and lactose by cancer cells, which increases the rate of glycolysis [99]. High expression of isoenzymes such as lactate dehydrogenase-A (LDHA) and lactate dehydrogenase-B (LDGB) in TNBC confirms the dependence of cells in this subtype of breast cancer on anaerobic glycolysis and correlates with a poor prognosis [100]. However, no separate studies have been conducted regarding the expression of the monocarbonate acid transporter family and a specific type of cancer. Only one study has performed a pan-cancer study on SLC16A10 expression. It has been shown that some members of the SLC16 family can be used for cancer prognosis and confirmed the association with immune invasion and the presence of a stem cell phenotype [101]. Particular attention is paid to tryptophan transport and the risk of developing hypertryptophanemia and tryptophan metabolism via the kynurenine pathway, which is immunosuppressive [102]. Studying the relationship between TNBC and the expression of SLC16 family transmembrane transporters is promising for understanding the metabolic features of TNBC and developing therapeutic targets.

The transmembrane transporter SLC33A1 is the only member of the acetyl-CoA transporter family [103]. In a recent study, a group of scientists found increased expression of acetyl-CoA synthetase 2 (ACSS2) in breast cancer. This enzyme catalyzes the reaction of converting acetate to acetyl-CoA. The reaction of acetyl-CoA synthesis is activated in response to nutrient deprivation and hypoxia [104]. Butylation was also shown to be a new class of histone modification, which involves non-acetyl acetylation. It is suggested that this reaction may promote the expression of genes responsible for breast cancer progression [105]. Acetyl-CoA is known to be a substrate for fatty acid synthesis [106].

In TNBC, one amino acid transporter from the proton-coupled amino acid transporter family, namely SLC36A1, exhibits increased activity. This transporter belongs to the neutral amino acid (proton) symporters. This transporter has pH-dependent electron transport activity for small amino acids (alanine, glycine, and proline). These amino acids potentiate mTORC1 signaling [107]. Initially, the association of SLC36A1 activity was shown in the study of kidney cancer. In this study, the functional axis TFE3-SLC36A1 was established and described. It is known that transcription factor E3 (TFE3) is a transcription factor involved in the activation of autophagic genes. For TFE3 activation, it is necessary for TFE3 to undergo the phosphorylation stage, after which it will be translocated. Phosphorylation of TFE3 regulates the availability of nutrients to the cell. It has been shown that during glucose starvation, TFE3 is activated by phosphorylation, which subsequently increases the activity of the amino acid transporter SLC36A1. SLC36A1 activity activates mTOR and cancer cell proliferation [108]. Subsequently, another group of scientists studied the functional properties of the folliculin tumor suppressor complex (FLCN) (FLCN, folliculin-interacting protein 1 (FNIP1), folliculin-interacting protein 2 (FNIP2)) and its contribution to the regulation of energy homeostasis through two kinases: AMP-activated protein kinase (AMPK) and mTORC1. According to the study, the suppression of the FLCN activity correlated with poor prognosis, especially in TNBC, in which the AMPK and TFE3 targets were activated. It was also found that FLCN-deficient cells with increased TFE3 activity activate the peroxisome proliferator-activated receptor-γ coactivator 1-α (PGC-1α)/HIF-1α pathways, resulting in the induction of aerobic glycolysis and angiogenesis [109].

In TNBC, the increased activity of amino acid transporters of the sodium-linked neutral amino acid transporter family in system A and system N SLC38A1, SLC38A2, and SLC38A3 is noted. A study of the expression level of SLC38A1 was conducted on the TNBC cell line MDA-MB-231, corresponding to the basally similar type B, which confirmed the association between TNBC and high levels of SLC38A expression [110]. Another study showed that SLC38A1 activity in TNBC correlates with high levels of Ki-67 proliferative activity, lack of estrogen receptor expression, metastases to regional lymph nodes, significant tumor size, and late stages of the disease [111]. The functional features of SLC38A2 will be described in more detail in the section on the adjacent expression of this transporter in TNBC and HER2-positive breast cancer. It is assumed that transporters 38A1 and 38A2 are functionally coupled. In-depth studies in this direction have not been conducted. Also, the expression of SLC38A3 via the glycogen synthase kinase-3 beta (GSK3β)/β-catenin/epithelial–mesenchymal transformation pathway promotes the metastasis of TNBC cells [112]. Little is known about the SLC38A4 transporter and its expression in TNBC. This transporter is a sodium-mediated amino acid transporter. It is involved in the transport of alanine, histidine, cysteine, asparagine, glycine, threonine, valine, glutamine, and methionine. It can mediate sodium-independent transport of cationic amino acids such as arginine and lysine [113].

## 6. Related Activity of Transmembrane Amino Acid Transporters in Different Molecular Biological Subtypes of Breast Cancer

The transporters SLC1A1 [110], SLC7A1, and SLC7A11 are highly expressed in luminal subtypes of breast cancer and TNBC. It is known that the transmembrane transporter SLC1A1 is a glutamate HIF-dependent transporter [114]. SLC1A1 is expressed in the luminal subtype of breast cancer, while SLC1A5 is expressed in the HER2-positive subtype. As mentioned earlier, SLC1A5 is involved in the transport of glutamine and asparagine. Glutamine acts as another signaling molecule for mTOR activation, which leads to cancer cell proliferation, suppression of apoptosis, and mediated autophagy, which is observed in TNBC [47]. In case of glutamine depletion or mitochondrial dysfunction, arginine is an essential amino acid for cancer cell survival [115]. Furthermore, high arginine levels in cancer cells are closely associated with epithelial to mesenchymal transition (EMT) and metastasis in breast cancer [116]. Here, we would like to suggest that the overall increased expression of SLC1A1 in TNBC and luminal breast cancer may be associated with the heterogeneity of the TNBC subgroup. It is known that when studying TNBC on cell lines, it is divided into two subgroups, basal-like A (with increased expression of estrogen receptors) and basal-like B (with no expression of estrogen receptors and increased proliferative activity of the HER2+ type). Apparently, the overall increased expression of SLC1A1 in luminal and TNBC is characteristic of basal-like A breast cancer. We can assume the same with the same expression of SLC1A5 with the positive expression of HER2 and TNBC. Increased activity may distinctively characterize basal-like B breast cancer. These assumptions are a hypothesis that requires practical confirmation. Also, in TNBC and luminal breast cancer subtypes, but not in those with positive HER2 expression, the adjacent expression of SLC7A1 and SLC7A11 is observed. This suggests that their expression is characteristic of luminal A and basal A-like breast cancer subtypes, where there is an increased expression of estrogen receptors. Such heterogeneity of TNBC is explained by the fact that cancer cells originate from two different cells of the mammary gland, namely luminal (inner) cells and basal (outer) cells. Luminal cells express luminal keratins 8, 18, 19, blocks programmed cell death 2 (BLC2), gene of mucin 1 (MUC1), and ER. In turn, basal cells, which exhibit features of epithelial and smooth muscle cells, express keratins 5, 14, and 17, smooth muscle actin, and p63 [117,118]. SLC7A11 functionally interacts with the transmembrane C-terminal subunit of MUC1 through binding to CD44v. The accumulation of glutamate in the extracellular space suppresses SLC7A11 activity in TNBC, which leads to a decrease in the entry of cysteine into the cell. Intracellular depletion of cysteine leads to the accumulation of ROS, which activates HIF-1α. Presumably, the activation of HIF-1α in TNBC occurs due to the glutamate efflux mechanism [119]. Transmembrane transporter SLC7A5 provides intracellular transport of large neutral amino acids (LNAA): leucine, isoleucine, tryptophan, tyrosine, methionine, and phenylalanine. As mentioned above, leucine is a universal and extremely important activator of mTOR. Apparently, this is why SLC7A5 is expressed in all subtypes of breast cancer. The increased expression of SLC7A5 is also associated with poor prognosis in patients with breast cancer [120].

In the luminal subtype and positive HER2 expression, the transmembrane transporters SLC7A7 and SLC7A8 have equally high expression activity (Figure 5). SLC7A7 and SLC7A8, as well as SLC7A1, SLC7A2, SLC7A5, are associated with the increased expression of hormone receptors and positive HER2 expression [78]. It has been shown that the SLC7A7 transporter is a target of PR [121]. SLC7A8 belongs to the Na + -independent, large neutral amino acid transporters 2. Several studies have shown the dependence of this receptor on the positive expression of HR and HER2, which suggests a specific expression of SLC7A8 in the luminal B subtype of breast cancer [122,123].

The increased activity of transmembrane transporters SLC1A5 and SLC38A2 was shown in TNBC and HER2-positive breast cancer. It was shown that the transmembrane amino acid transporter SLC1A5 of the high-affinity glutamate and neutral amino acid transporter family exhibits increased expression in the HER2+ subtype and TNBC. It transports mainly alanine, serine, glutamine, cysteine, threonine, and asparagine [124]. Cancer cells with a positive HER2 expression of breast cancer also demonstrate active glutamine consumption. This is evidenced by the high activity of GLS1, glutamine dehydrogenase (GDH), and the SLC1A5 glutamine transporter Alanine–Serine–Cysteine Transporter (ASCT2), which also transports alanine, serine, and cysteine [125,126]. The same active expression of SLC1A5 ASCT2 was shown earlier when describing luminal subtypes of breast cancer. With the luminal B subtype and positive expression of HER2, the MYC oncogene is activated, which significantly increases the expression of the transmembrane amino acid transporter SLC1A5 ASCT2 and causes crosstalk between ER and HER [127]. Due to the influx of glutamine into the cell, the mTOR pathway is activated, which triggers cell growth and proliferation in response to the influx of nutrients into the cell [121]. In one study, it was shown that the affinity of SLC1A5 for glutamine increases at low pH. Since the pH of a tumor cell with a more aggressive phenotype has a low pH compared to a healthy one, the penetration and consumption of glutamine occurs more often and faster in tumor cells [128]. SLC1A5 activity is regulated by c-MYC and mutp53. Missense mutations in MUTP53 have been shown to result in SLC1A5 and SLC1A4 activation, as well as phosphoenolpyruvate carboxykinase 2 (PCK2), which enhances serine production [43]. Serine is involved in gluconeogenesis as well as in the redox balance of the tumor cell [41]. In one of the studies, scientists discussed the affinity and the need for expression coupling between some transmembrane amino acid transporters. One study reported that with reduced functionality of SLC1A5 and SLC7A5, glutamine continued to enter the cell through SLC38A1 and SLC38A2 [83]. This study reveals the reasons for such resistance of cancer cells when some amino acid transporters are switched off. There are many collaterals and bypass routes for amino acid transport into cells. SLC1A5 has also been shown to be associated with endocrine resistance in breast cancer cells. Another common feature between luminal breast cancer subtypes and positive HER2 expression is the activation of IGF-R1. In the case of the HER2-positive subtype of breast cancer, heterodimerization of HER2 and IGF-R1 occurs through high levels of truncated isoforms of dopamine and c-AMP-regulated phosphoprotein (t-DARPP) [129,130]. As mentioned earlier, IGF itself and its receptor IGF-R1 are estrogen targets. There is also a similarity in the metabolism of TNBC cells and cells with positive HER2 expression. Using immunohistochemical staining of breast cancer tissues, an increased expression of proteins responsible for glutamine metabolism was shown to be greater among HER2-positive and TNBC than in other subtypes [131]. The increased expression of SLC38A1 correlated with AKT phosphorylation, which enhanced the proliferative potential of tumor cells [108]. A high expression of SLC38A2 has been reported in both HER2-positive and TNBC, which most likely corresponds to the basal type B TNBC. The functional features of SLC38A2 are described in the section on HER2-positive breast cancer and will not be discussed in detail here. The SLC38A3 transporter showed the most specific and pronounced expression in TNBC. SLC38A3 is involved in the transport of glutamine, glutamate, asparagine, aspartate, alanine, and glutathione (GSH) in TNBC cells [104].

The expression of the transmembrane amino acid transporter SLC7A5 is active in all molecular biological subtypes of breast cancer [132]. The transmembrane transporter SLC7A5 provides intracellular transport of large neutral amino acids (LNAA): leucine, isoleucine, tryptophan, tyrosine, methionine, and phenylalanine. As mentioned above, leucine is a universal and extremely important activator of mTOR. Apparently, this is why SLC7A5 is expressed in all subtypes of breast cancer. The increased expression of SLC7A5 is also associated with poor prognosis in patients with breast cancer [121]. One study reported that SLC7A5 potentiated the proliferation of luminal breast cancer cells by activating AKT/mTORC1 through phosphorylation [133]. Another source from the literature reported that SLC7A5 is associated with an aggressive luminal subtype of breast cancer characterized by high proliferative activity controlled by MYC activity. Thus, a high expression of SLC7A5 mRNA and protein were closely associated with poor prognosis, large tumor size, and high grade of malignancy. A high expression of SLC7A5 was shown in TNBC, positive HER2 expression, and the luminal B subtype [134].

## 7. Potentially New Transmembrane Amino Acid Transporters for Study in Breast Cancer

In addition to the listed transmembrane amino acid transporters, which have been studied to varying degrees in breast cancer, there are still a very large number of transporters that require further study. In some cases, only the basic properties of amino acid transporters are described without considering their functional activity depending on the pathology. In other cases, their activity is considered only in one type of benign or malignant pathology. Since some pathologies have a similar mechanism of activation in signaling pathways and have a similar metabolic profile, then, in our opinion, it is considered appropriate to deepen our knowledge and understanding of their functional activity and ability in different types of malignant neoplasms, including breast cancer.

Of the high-affinity glutamate and neutral amino acid transporters SLC6 family, 21 amino acid transporters are known (Figure 6). However, only four of this family have been studied in breast cancer. The increased activity of these transporters has been shown in TNBC as well as in luminal subtypes. This family has affinity for a wide range of amino acids [135,136,137,138]. The most studied family of amino acid transporters in breast cancer is the cationic amino acid transporter/glycoprotein-associated family SLC7. Only three members of the family have not been described in breast cancer, two of which have not been studied for amino acid affinity.

To date, only four members of the family are known, with only one, SLC36A1, well characterized in TNBC. The SLC36 family is known as proton-coupled amino acid transporters and functions as sodium-dependent amino acid transporters. It belongs to the pH-sensitive amino acid cotransporters. SLC36A32 is known to mediate the electrogenic cotransport of glutamine and sodium ions in exchange for protons. In addition, it is involved in the transport of histidine, alanine, and asparagine. The affinity of the SLC36A3 transporter for amino acids has not been identified. The transport of amino acids into the cell via SLC36A34 differs from all other members of the SLC36 family. Thus, SLC36A4 has the highest affinity for tryptophan and proline. To a lesser extent, it is involved in the transport of alanine and is completely intact with glycine and cysteine [104,139].

The family of neutral amino acids associated with sodium in system A and system N SLC38 has been identified, with about 11 members of the family, 5 of which have been well studied in TNBC and the positive expression of HER2 on the surface of cancer cells. In other cases, it has not been established which amino acids are transported by the remaining members of this family.

The activity of the Na-independent, systemic L-like family of amino acid transporters SLC43, mitochondrial pyruvate transporters SLC54, and PQ-loop amino acid transporters SLC66 in breast cancer has not been studied at all. Little is known about the fact that the SLC43 family includes sodium-dependent transporters with a high affinity for large neutral amino acids (leucine, isoleucine, valine, phenylalanine, and methionine) [140,141]. It has also been shown that SLC43A1 is more selective with phenylalanine and branched-chain amino acids compared to SLC7A5 and SLC7A8 [142].

## 8. Relationship Between Amino Acid Metabolism and Immune Regulation in Different Molecular Biological Subtypes of Breast Cancer

The tumor does not exist in an isolated state but actively interacts with its microenvironment and affects the state of the immune system. We examined the relationship between amino acid metabolism and the functional state of the immune system in breast cancer.

The group of anionic amino acids includes Glu and Asp. The main role of these amino acids is the synthesis of pyrimidine nucleotide bases, which are necessary for cell division, including immune cells [143]. Asp is formed into Glu via the enzyme aspartate aminotransferase (GOT1) and maintains the metabolism of the mitochondrial matrix by transferring reduced equivalents of the electron transport chain (ETC) [144]. Asp also participates in the production of plasma cells. Glu, converted into α-KG acid, feeds the TCA, activates M1 type macrophages (pro-inflammatory type), IL-1β, and methylates histones. Like Asp, it affects the production of plasma cells [145]. It is also part of GSH, implementing anti-inflammatory functions [146].

The cationic amino acids group includes His, Lys, and Arg. The presence of free extracellular Arg is necessary for the production and proliferation of natural killer (NK) cells. Arg metabolism can follow a pro-inflammatory pathway with the formation of M1 macrophages under the influence of inducible isoform nitric oxide synthase (iNOS). Citrulline and nitric oxide (NO) are formed, which suppress OXPHOS, activating glycolysis and M1 type macrophages. Also, NO leads to the loss of the FeS cluster, the accumulation of Fe, and cell ferroptosis [147]. Under the action of arginase I, Arg is converted to ornithine. Further, under the action of the enzyme ornithine decarboxylase (ODC), ornithine is converted to the polyamine spermine. Spermine forms hypusine, which is necessary for the activation of the eukaryotic translation initiation factor 5A (eIF5a) [148]. The eIF5a factor is responsible for the elongation and termination of translation, activates ETC and OXPHOS in M2 and IL-4 [149,150]. Arg is involved in the regulation of the T-cell division cycle in the G1 phase through cyclin-dependent kinase 4—Cyclin3 (CDK4/Cyclin3) [151]. Arg is also involved in DNA methylation, and therefore regulates T-cell differentiation and cytokine secretion [152]. The role of His and Lys in immune cell metabolism has not been sufficiently described. It is known that these amino acids feed the TCA, the main source of energy for cells, including immune cells. Thus, Lys is incorporated into the TCA via acetoacetyl-CoA, and His via glutamate [153].

The large neutral amino acids group includes Leu, Ile, Met, Phr, Tyr, Trp. Thus, Leu and Ile are branched-chain amino acids (BCAA). Leu is able to accumulate in lysosomes, is a key activator of mTORC1, and regulates proliferation and differentiation of T cells and Treg cells [154,155]. Met is necessary for DNA methylation, which determines the proliferation of T cells and the secretion of cytokines. In addition, it inhibits Treg cell apoptosis. Met also activates S-Adenosylmethionine (SAM), which suppresses the antitumor effects of T cells [156,157]. The amino acids Phe and Tyr provide immune cells with energy by feeding the TCA. Phe enters the TCA due to acetoacetyl-CoA and fumarate, while Tyr enters due to acetoacetyl-CoA, fumarate, and pyruvate [153]. Trp also feeds the TCA via acetoacetyl-CoA, acetyl-CoA, and pyruvate via general control nonderepressible 2 (GCN2) and exerts immune suppression, activates programmed cell death protein 1 (PD-1), and regulates T cell proliferation, in general, and Treg cell activity in particular [153].

The small neutral amino acid group includes Ala, Ser, Cys, Thr, Pro, Asn, Gly. Thus, Ala is involved in mitochondrial metabolism and mitogenic activity of CD4+. Also, Ala activates naive T cells [158]. Ala, through the formation of pyruvate, activates PD-1L, feeds the TCA and modifies collagen. Collagen modification simplifies cell migration, including cancer cells [159]. Sufficient levels of Ser activate T cell proliferation through serine hydroxymethyltransferase-2 (SHMT2) [160]. It is part of GSH and affects the activity of immunosuppressive Treg cells. Also, Ser is involved in the activation of pro-inflammatory signals through the activation of pyruvate kinase M2 (PKM2) [161]. As a result, phosphoenolpyruvate is converted into pyruvate, glycolysis, and activation of M1 type macrophages, TNF, IL-1β, and IL-17 are triggered [162]. The deficiency of Ser leads to destabilization of the FeS cluster, accumulation of Fe inside the cell, and subsequent ferroptosis [163]. The amino acid Cys also participates in maintaining the FeS cluster, as does Ser [164]. It is part of GSH, due to which anti-inflammatory functions are realized [165].

Cys also affects DNA synthesis, the proliferation of T and B cells and their activation [166,167]. The amino acid Thr provides immune cells with energy by maintaining the TCA through succinyl-CoA and pyruvate [153]. It has been shown that Pro, together with pyruvate, participate in the modification of collagen, promoting metastasis of cancer cells [168]. It has been shown that Asn regulates the activity of CD8+ cells [169]. Through the formation of Gln and then Leu, it activates mTORC1, which leads to accelerated division and differentiation of T cells and Treg cells [170]. The amino acid Gly is part of GSH and activates T and B cells, NK, macrophages, and IL-1β [171,172]. It is part of purine nucleotide bases [173]. Gly, via SHMT2, is involved in stimulating Ser metabolism, which also leads to stimulation of cell proliferation, including immune cells [174]. In addition, Gly affects mitochondrial metabolism and mitogenic activity of CD8+ [175].

The all neutral amino acids group includes Leu, Ile, Val, Asn, Gln, Gly, Cys, Met, Phr, Pro, Ser, Thr, Trp, and Tyr. The functions of all amino acids (except Val and Gln) and their role in regulating the immune system were listed above. Thus, Val is known for being a BCAA, as are Leu and Ile. Basically, the functions of Val are not considered separately but in the group of all BCAA. It is known that all BCAA amino acids participate in the nutrition of the TCA through acetyl-CoA, succinyl-CoA, and α-KG acid [176]. All BCAA amino acids participate in the acetylation of histones due to the HDAC enzyme [177]. BCAA also acetylate NF-kB, the nucleotide-binding and oligomerization domain (NOD), leucine-rich repeat (LRR), and pyrin domain-containing protein 3 (NLRP3), which activate glycolysis and pro-inflammatory activation of the immune system [178,179]. The role of Gln is diverse. The amino acid is a substrate for the formation of GSH, activating mTORC1. Through the formation of α-KG acid, it activates M1 type macrophages, feeds the TCA, provides immune cells with energy, and is a cofactor of histone demethylases [180]. Gln activates the production and proliferation of NK and B cells [181]. It is also part of the hexosamine biosynthesis pathway (HBP). It forms uridine diphosphate-N-acetylglucosamine (UDP-GlcNAc), which is involved in the glycosylation of CD4+ and CD8+ cells, providing their intercellular signaling [182,183]. UDP-GlcNAc acts as a substrate for the formation of O-GlcNAcT (OGT), which is involved in the renewal of thymus T cells, the production, activation and expansion of T cells, the activation of IL-2, and RNA synthesis [184,185,186].

In luminal subtypes of breast cancer, the expression level of proteins of transporters carrying Asp, Glu, Ala, Arg, BAAC, and Met is increased. The set of these amino acids reflects that the body undergoes active proliferation, differentiation, and expansion of the cellular profile due to the active metabolism of Asp (synthesis of pyrimidines), Glu, Ala (TCA), BCAA, and Met (regulation of DNA gene activity due to acetylation and methylation) as well as controlled pro-inflammatory activity due to the presence of both pro-inflammatory and anti-inflammatory amino acids (Arg, Glu, BCAA). The HER2+ subtype of breast cancer has an excellent metabolic profile. Cancer cells metabolize in even greater quantities of amino acids that describe the activity of proliferative processes (Ser, Cys, Gln, BAAC, Gly), the remodeling of collagen fibers for the growth and metastasis of tumor cells (Ala), and the synthesizing of fatty acids as distinctive features of the HER2+ subtype of breast cancer (Cys), which has more suppressed immunity (Trp). TNBC has a completely different metabolic profile in terms of the remodeling of collagen fibers (pyruvate, Pro) and glycolytic activity (lactate, Gln, acetyl-CoA). In all other respects, the metabolic profile of TNBC is largely comparable to HER2+.

If we consider the relationship between the activity of transmembrane amino acid carriers in cancer and immune cells, their mechanism of communication is not fully understood. We assume that cancer cells can completely change the metabolism of immune cells in the tumor microenvironment by adjusting it in an identical way to their metabolism. Also, it is possible that cancer and immune cells will have a mutually opposite mechanism of amino acid exchange. Thus, there may be a competitive struggle for amino acids between cancer and immune cells. In this case, cancer cells can reprogram the metabolism of tumor-associated immune cells and other nearby cells in such a way that their vital activity and functioning will be aimed at ensuring the most favorable state of cancer cells to the detriment of their own needs. No studies have been conducted regarding the presence or absence of symmetrical activation of transmembrane amino acid carriers in both cancer and immune cells. It may turn out that the activity of certain amino acid transmembrane transporters in breast cancer cells of a certain phenotype individually activates cytokines that will activate completely different transmembrane amino acid transporters.

## 9. Genetics and Epigenetics of Transmembrane Amino Acid Transporters Depending on the Molecular Biological Subtype of Breast Cancer

To analyze the genetic and epigenetic regulation of transmembrane amino acid transporters, we used the open GeneCards database. We collected information on the genes encoding transmembrane amino acid transporters, their locations in chromosomal loci, and the latest data on miRNAs that affect protein expression. The expression of one transmembrane transporter can be regulated by one to two hundred miRNAs (Appendix A). Moreover, the expression of one miRNA can act on an amino acid transporter in various cells in breast cancer: in breast cancer cells themselves, in tumor microenvironment cells, in unchanged and tumor-associated immune cells. This once again emphasizes the complexity of the metabolic network. In this review, we limited ourselves to only generalizing the latest information on the genetics and epigenetics of transmembrane amino acid transporters in relation to breast cancer phenotypes. A thorough study of the complex and intertwined connections of metabolic pathways in relation to amino acids in breast cancer is a topic for a separate theoretical work. We assume that the information collected in this section will be useful for future practical studies of the regulation of amino acid metabolism at the epigenetic level in breast cancer cells with different phenotypes.

## 10. Promising Strategies for Breast Cancer Treatment Based on the Characteristics of Transmembrane Amino Acid Transporters

Understanding the mechanisms of operation of transmembrane amino acid carriers will allow us to optimize and develop a more effective approach to treating patients with breast cancer with chemoresistance in the future. It will also help us to select a more effective approach to treating patients with the most complex subtypes. One option could be the pharmacological induction of some transmembrane transporters, which are antagonists of another target transporter [187].

It is also possible to regulate miRNA expression, thereby influencing protein expression. Based on the little data that exists on the interaction of miRNA and transporter protein, most of them are inversely proportional to activity [67,72,188,189,190].

Genes of amino acid transporter proteins are also target points. It is possible to influence GATA2 through activation of the PTEN/AKT signaling pathway [68]. By activating protein phosphatase 2A (PP2A), it is possible to suppress c-Myc expression [191]. Potential routes to pursue in breast cancer therapy are the inhibition of the SAM-MAT2A [192] and SAM-mTORC1 [193] axes, as well as suppression of the expression of GCN2 [194], MAPK, ERK, JNK [195,196], HIF [197], and NF-kB [198]. Several therapeutic approaches have been developed for breast cancer with mutant and wild-type p53 [199]. It has been shown that suppression of mTORC1 and downstream S6K1 and 4E-BP can be an effective therapeutic target for patients with breast cancer [200]. Here, we have only partially shown possible therapeutic targets without going into details, as this is the topic of a separate literature review.

## 11. Limitations

The limitations of this review include the analysis based on cell lines rather than tissue biopsies. It is important to consider that the microenvironment consisting of immune cells, tumor-associated immune cells, fibroblasts, and adjacent tissue cells of native cancer cells significantly contribute to the metabolic profile that we can observe in biopsies.

Also, we did not delve into the mechanisms of miRNA influence on the expression-related transmembrane amino acid transporters in breast cancer. There are no studies on the relationship between the amino acid metabolic landscape of breast cancer cells of each individual subtype and the immune cells that are part of the tumor microenvironment.

Another limitation is the lack of a detailed description of the mechanisms of action of some transmembrane amino acid transporters. To date, only individual members of the transmembrane amino acid transporter families, especially the SLC1 and SLC7 families, have been well studied. Research is needed on the remaining transporters.

## 12. Conclusions

This comprehensive review examined amino acid metabolism in various molecular biological subtypes of breast cancer in terms of the activity of all currently known transmembrane amino acid transporters. We show that the metabolism occurring in cancer cells cannot be considered solely in one perspective of amino acid consumption, promoting their growth, development, and proliferation. It is a complex system that is regulated at the epigenetic, genetic, hormonal, and immunological levels, as well as through the activity of signaling pathways.

It is evident that the main signaling pathway responding to metabolic shifts in the cell is PI3K/Akt/mTOR. In addition, the activity of the oncogene c-Myc, HIF-1α, p53, GATA2, NF-kB and MAT2A has a direct effect on the amino acid metabolism of cancer cells, their growth and proliferation, as well as maintaining homeostatic balance. At the same time, a distinctive feature of luminal subtypes of breast cancer from TNBC is the ability for gluconeogenesis. It can be assumed that in HER2+ and HR+, mTOR activation is carried out through leucine, while in TNBC, mTOR signaling occurs through arginine and glutamate, which is converted into glutamine under the action of the enzyme glutamine synthetase. HER2 receptor-positive breast cancer, unlike TNBC and the luminal A subtype, has a distinctive active synthesis and consumption of fatty acids. Another observation is that pyruvate and glutamine are common metabolites for the luminal A and B subtypes of breast cancer. At the same time, the implementation of pyruvate and glutamine occurs differently. In luminal A, the active consumption of pyruvate and glutamine is associated with gluconeogenesis. In the luminal B subtype with high proliferative activity, the absorption of pyruvate and glutamine is driven by the active process of lipogenesis. Interestingly, amino acid transporters exhibit their activity depending on the pH level inside the cell. In the most aggressive forms of breast cancer or with gradual progression of the disease, pH will also change, which will directly affect the metabolism of amino acids. Using the cell lines presented in this review, we can trace the characteristic features inherent in each of the molecular biological subtypes of breast cancer and develop the most optimal therapeutic targets.

This work reviews the features of amino acid metabolism in all breast cancer subtypes through the activity of transmembrane amino acid transporters. Information on the genetic and epigenetic regulation of transporter proteins is provided. The relationship between immune regulation and amino acid metabolism in breast cancer is described. Potential therapeutic targets are identified. Still, the properties of some amino acid transporters and their role in breast cancer remain unexplored. There are no studies on the relationship between amino acid metabolism of tumor-associated immune cells and breast cancer cells in different subtypes. Also, there are also no studies on the relationship between the expression of all possible microRNAs associated with amino acid transporter protein genes and the expression of these transporter proteins. This review provides a comprehensive understanding of the complex nature of the molecular biology of breast cancer subtypes. It identifies several relevant directions for future research. Finding answers to these questions will allow us to form a better understanding of the various molecular biology subtypes of breast cancer and solve problems associated with the chemoresistance of cancer cells.

## Figures and Tables

**Figure 1 cimb-47-00004-f001:**
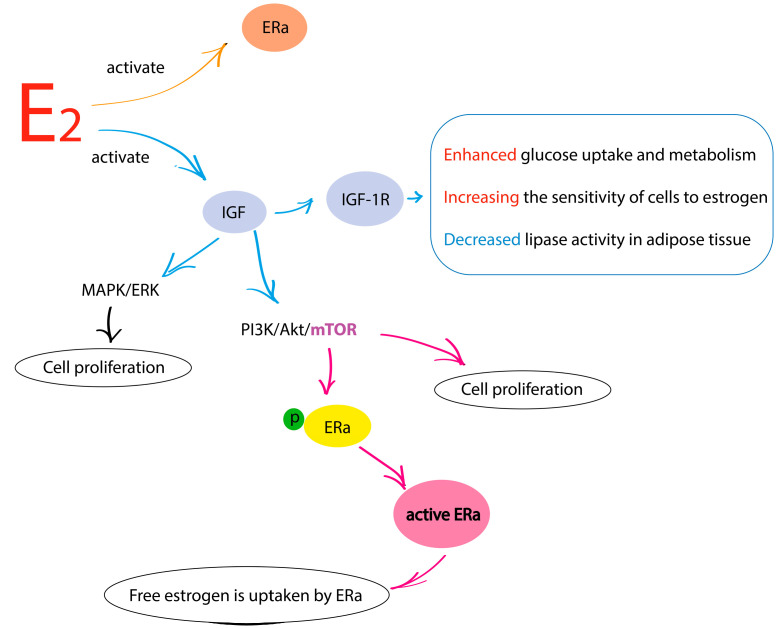
Estrogen targets. E_2_-estrogen; ERa—estrogen receptor a; IGF—insulin-like growth factor; IGF-1R—insulin-like growth factor 1 receptor; MARK/ERK—mitogen-activated protein kinase/extracellular signal-regulated kinases; PI3K/Akt/mTOR—phosphoinositide 3-kinases/protein kinase B/mammalian target of rapamycin.

**Figure 2 cimb-47-00004-f002:**
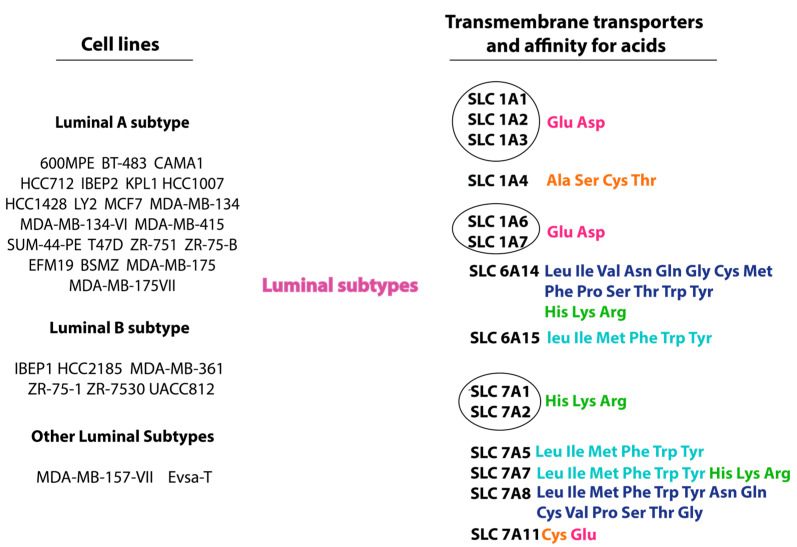
Cell lines, transmembrane transporters, and amino acids related to luminal subtypes of breast cancer. The circled group represents a group of transmembrane amino acid transporters with a common affinity for amino acids. *Anionic amino acids*: Glu, Asp; *small neutral amino acids*: Ala, Ser, Cys, Thr; *all neutral amino acids*: Leu, Ile, Val, Asn, Gln, Gly, Cys, Met, Phe, Pro, Ser, Thr, Trp, Tyr; *cationic amino acids:* His, Lys, Arg; *large neutral amino acids:* Leu, Ile, Met, Phe, Trp, Tyr.

**Figure 3 cimb-47-00004-f003:**
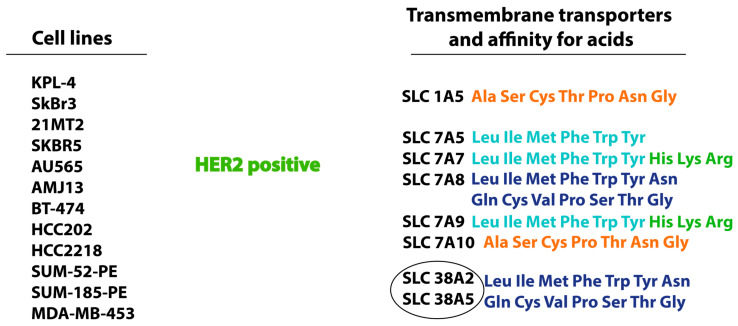
Cell lines, transmembrane transporters, and amino acids related to the HER2 positive subtype of breast cancer. The circled group represents a group of transmembrane amino acid transporters with a common affinity for amino acids. *Anionic amino acids*: Glu, Asp; *small neutral amino acids*: Ala, Ser, Cys, Thr; *all neutral amino acids*: Leu, Ile, Val, Asn, Gln, Gly, Cys, Met, Phe, Pro, Ser, Thr, Trp, Tyr; *cationic amino acids:* His, Lys, Arg; *large neutral amino acids:* Leu, Ile, Met, Phe, Trp, Tyr.

**Figure 4 cimb-47-00004-f004:**
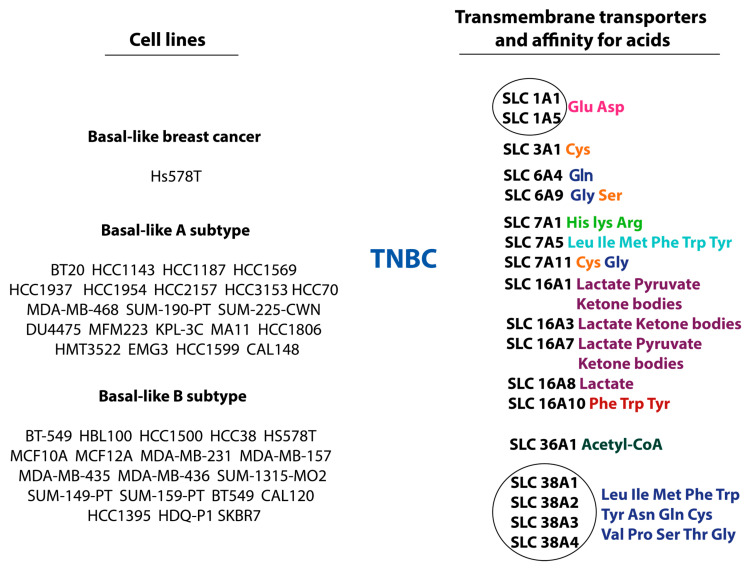
Cell lines, transmembrane transporters, and amino acids related to the HER2 positive subtype of breast cancer. The circled group represents a group of transmembrane amino acid transporters with a common affinity for amino acids. *Anionic amino acids*: Glu, Asp; *small neutral amino acids*: Ala, Ser, Cys, Thr; *all neutral amino acids*: Leu, Ile, Val, Asn, Gln, Gly, Cys, Met, Phe, Pro, Ser, Thr, Trp, Tyr; *cationic amino acids:* His, Lys, Arg; *large neutral amino acids:* Leu, Ile, Met, Phe, Trp, Tyr; *aromatic amino acids:* Phe, Trp, Tyr; *monocarboxylate:* lactate, pyruvate, ketone bodies.

**Figure 5 cimb-47-00004-f005:**
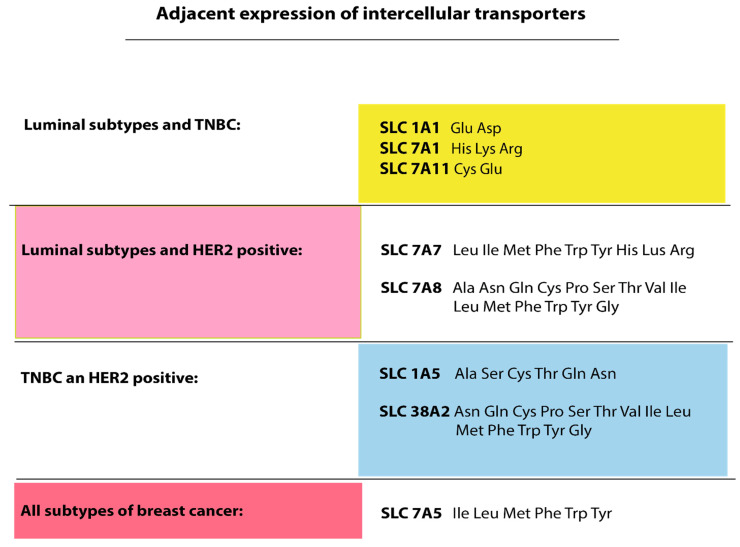
Adjacent expression of intercellular transporters among subtypes of breast cancer.

**Figure 6 cimb-47-00004-f006:**
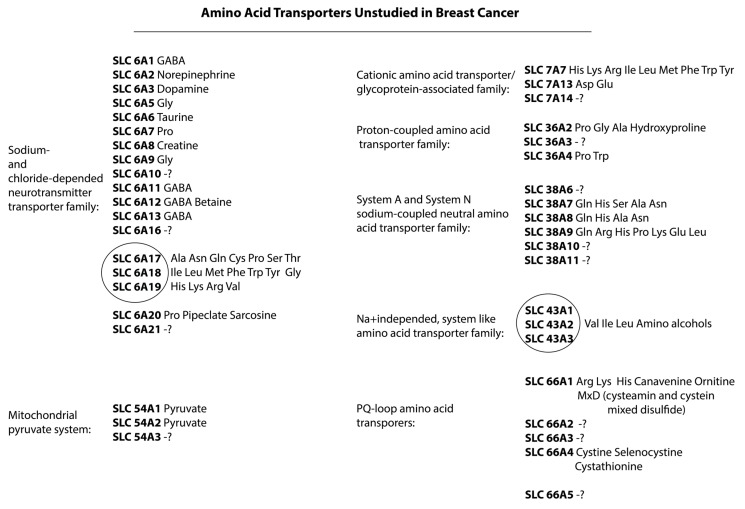
Amino acid transporters unstudied in breast cancer. The circled group represents a group of transmembrane amino acid transporters with a common affinity for amino acids. The sign «?» indicates those transporters whose affinity for amino acids has not yet been studied.

## Data Availability

Not applicable.

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
