# Peer review of "Transmembrane Amino Acid Transporters in Shaping the Metabolic Profile of Breast Cancer Cell Lines: The Focus on Molecular Biological Subtype"

_cimb, 2024, doi:10.3390/cimb47010004_

Round 1
Reviewer 1 Report
Comments and Suggestions for Authors
The manuscript of Elena I. Dyachenko et. al presents a relatively comprehensive review on the main classes of amino acids transporters and their physiological roles in the metabolic activity linked to the survival of cancer cells. The review is highlighting the changes in the amino acid composition of different subtypes of breast cancer cells and the relationship between the changes in the pH and the activity of amino acid transporters.
Moreover, the authors present the signaling pathways mediating the activity of the amino acid transporters which are regulated by the signal transduction events mediated by other well-known cancer-related pathways such as main signaling pathway (PI3K/Akt/mTOR), the activity of the oncogene c-Myc, HIF, p53, GATA2, NF-kB and MAT2A, altogether contributing to the cancer cells growth and proliferation, as well as the maintenance of their homeostatic equilibrium.
The manuscript is well written, minor changes in English language in the last paragraph in the discussion section is required.
Some minor changes are suggested for increasing the exposure to different audiences in the field of breast cancer:
1. The author should better define the roles of some amino acid transporters in inducing different chemoresistance phenotypes: which are the breast cancer subtypes which showed specific mutations, deletions in the genes of such amino acid transporters and had specific outcome in the survival rate in response to different chemotherapies and or immunotherapies/radiation treatment?
2. The authors should show a table with the main amino acid transporters identified as pharmacological targets of interest for the chemotherapeutics approaches involved in the treatment of the breast cancer subtypes mentioned in this review.
Author Response
Reviewer 1
The authors are grateful to the reviewers for careful consideration of the manuscript and valuable comments. We hope that thanks to the joint work, the manuscript has become better.
The manuscript of Elena I. Dyachenko et. al presents a relatively comprehensive review on the main classes of amino acids transporters and their physiological roles in the metabolic activity linked to the survival of cancer cells. The review is highlighting the changes in the amino acid composition of different subtypes of breast cancer cells and the relationship between the changes in the pH and the activity of amino acid transporters.
Moreover, the authors present the signaling pathways mediating the activity of the amino acid transporters which are regulated by the signal transduction events mediated by other well-known cancer-related pathways such as main signaling pathway (PI3K/Akt/mTOR), the activity of the oncogene c-Myc, HIF, p53, GATA2, NF-kB and MAT2A, altogether contributing to the cancer cells growth and proliferation, as well as the maintenance of their homeostatic equilibrium.
The manuscript is well written, minor changes in English language in the last paragraph in the discussion section is required.
Some minor changes are suggested for increasing the exposure to different audiences in the field of breast cancer:
- The author should better define the roles of some amino acid transporters in inducing different chemoresistance phenotypes: which are the breast cancer subtypes which showed specific mutations, deletions in the genes of such amino acid transporters and had specific outcome in the survival rate in response to different chemotherapies and or immunotherapies/radiation treatment?
Thank you for your comment. We have added the relevant information to the new therapy section.
- The authors should show a table with the main amino acid transporters identified as pharmacological targets of interest for the chemotherapeutics approaches involved in the treatment of the breast cancer subtypes mentioned in this review.
A complete and detailed list of specific amino acid transporters as therapeutic targets is an extensive topic. This is the topic of a separate literature review. In the section on therapy, we outlined general possible therapeutic targets and provided several examples.
Reviewer 2 Report
Comments and Suggestions for Authors
According to the editor’s strict regulation, I have carefully read and checked the review article described by Elena I. Dyachenko and Lyudmila V. Bel’skaya based on its scientific significance, soundness and novelty.
1. Overall evaluation
This review focuses on the role of transmembrane amino acid transporters in shaping the metabolic characteristics of breast cancer cell lines, with a focus on molecular biology subtypes, which is of great research significance. The paper covers a wealth of content and provides a comprehensive description of amino acid metabolism and related transport proteins in different subtypes of breast cancer, providing a valuable reference for a deeper understanding of the metabolic mechanisms of breast cancer. However, there is still room for improvement in some areas, as detailed below.
2. Specific problems and suggestions for improvement
(1) Completeness of literature references and reviews
Some research results are mentioned in part of the text, but the relevant literature is not clearly cited. For example, when describing the function of some amino acid transport proteins, there is a lack of corresponding literature support, which affects the credibility of the conclusions to a certain extent.
For some emerging or controversial ideas, there is insufficient comprehensive discussion with full citations from different research directions. For example, when discussing the relationship between certain transport proteins and breast cancer prognosis, only some supporting studies are cited, and no mention is made of the possible opposite views or the influence of other related factors.
(2) Research on innovation and depth
Although a relatively comprehensive summary has been made of the known roles of transmembrane amino acid transporters in breast cancer, it is lacking in terms of innovation. Most of the content is a consolidation of existing research results, and there is a lack of in-depth exploration of new discoveries or unique insights.
The potential mechanisms of some transport proteins in breast cancer metabolism are not explained in sufficient depth. For example, when discussing the interaction between transport proteins and signal pathways, only a superficial description of the association is provided, without exploring in depth the regulatory mechanism and dynamic process at the molecular level.
(3) Logical structure and clarity of expression
The overall logical structure of the article is relatively clear, but in some chapters, the organisation of the content is slightly loose. For example, when introducing the metabolic characteristics and transport proteins of different subtypes of breast cancer, there is some repetition of expressions and scattered information, which affects reading fluency.
The use of some technical terms and abbreviations is not fully explained, which may cause difficulties for readers who are not experts in the field. For example, the names of some signal pathways (such as PI3K/Akt/mTOR) and transporter families (such as SLC6) that frequently appear in the text should be fully explained when they first appear.
(4) Graph and chart quality and data analysis
The graphs and charts provided in the text (e.g., Figs. 2–5) help visually demonstrate the relationship between cell lines, transmembrane transport proteins, and amino acids. However, the quality and labeling of the graphs and charts could be improved. For example, the font in some graphs and charts is small, and the lines and markers are not clear enough, which affects the communication of information.
There is a lack of in-depth analysis and interpretation of the data in the figures. The correspondence between transport proteins and cell lines is simply listed, and the biological significance and potential laws behind the data are not further explored. Statistical methods can be used to analyse the data, such as comparing whether the differences in the expression levels of transport proteins in different subtypes are statistically significant, and the results of the analysis should be elaborated in the text.
(5) Discussion of the clinical application value of the research
Although the paper mentions the significance of the research results for the development of treatment targets, there is a lack of specific discussion and prospects on how to translate these research findings into actual clinical applications. For example, there is no detailed explanation of how to design more effective treatment strategies for breast cancer based on the characteristics of transport proteins, as well as the challenges and solutions that may be faced in clinical practice.
(6) Conclusion section improvement
Although the conclusion section summarises the main content of the research, it lacks a high-level summary and sublimation of the research results. It fails to highlight the important contribution of the research in the field of breast cancer research and provide a clear guide for future research directions.
(7) Language and grammar issues
There are some places in the text where the language is not precise or fluent enough, and some sentences have complex structures, which affect the reader's understanding. For example, there are grammatical errors or unclear logic in some long sentences.
There are a few problems with inappropriate word choice, such as the use of some technical terms that are not precise enough or inappropriate in a particular context.
(8) Discussion of the limitations of cell line research
The limitations of using cell lines as a research tool were not adequately discussed. Cell line research may not fully simulate the complex environment of tumours in vivo, which may affect the extrapolation of research results, but this was not mentioned in the article.
(10) Timeliness of data updates
Some of the cited literature is relatively old and may not reflect the latest research developments in the field. In the rapidly developing field of biomedicine, the timeliness of data is crucial to the quality of the review.
(11) Systematic research on the function of transmembrane transport proteins
There is a lack of systematic integration and analysis of the function of some transmembrane transport proteins. For example, when introducing different families of transport proteins, their role in different subtypes of breast cancer is described separately, but the synergistic or antagonistic effects of these transport proteins in the metabolic network of breast cancer are not discussed in depth as a whole.
(12) An in-depth discussion of the relationship between breast cancer heterogeneity and the expression of transport proteins
Although the heterogeneity of breast cancer and the differences in the expression of transport proteins are mentioned in the article, there is a lack of in-depth discussion of the intrinsic connection and molecular mechanism between this heterogeneity and the expression of transport proteins. For example, it is not elaborated in detail how the differences in the genetic background and epigenetic regulation of breast cancer cells of different subtypes affect the expression and function of transport proteins.
(13) Expanding the relationship between immune regulation and amino acid metabolism
With the development of tumor immunotherapy, the role of immune regulation in the treatment of breast cancer has received increasing attention. However, the relationship between amino acid metabolism and immune regulation is not discussed in the paper. Amino acid metabolism not only affects the growth and proliferation of cancer cells, but may also affect the function of immune cells in the tumor microenvironment. However, this is not fully discussed in the paper.
(14) Integration of the regulation of transport proteins by non-coding RNAs
Non-coding RNAs play an important role in the regulation of gene expression, but the paper does not systematically integrate the regulatory effects of non-coding RNAs (such as miRNAs and lncRNAs) on transmembrane amino acid transport proteins. Existing studies have shown that non-coding RNAs can regulate the expression of transport proteins through various mechanisms, affecting the metabolism of breast cancer cells, but these contents are not fully reflected in the paper.
Comments on the Quality of English LanguageThere are some places in the text where the language is not precise or fluent enough, and some sentences have complex structures, which affect the reader's understanding. For example, there are grammatical errors or unclear logic in some long sentences.
There are a few problems with inappropriate word choice, such as the use of some technical terms that are not precise enough or inappropriate in a particular context.
Author Response
Reviewer 2
The authors are grateful to the reviewers for careful consideration of the manuscript and valuable comments. We hope that thanks to the joint work, the manuscript has become better.
According to the editor’s strict regulation, I have carefully read and checked the review article described by Elena I. Dyachenko and Lyudmila V. Bel’skaya based on its scientific significance, soundness and novelty.
- Overall evaluation
This review focuses on the role of transmembrane amino acid transporters in shaping the metabolic characteristics of breast cancer cell lines, with a focus on molecular biology subtypes, which is of great research significance. The paper covers a wealth of content and provides a comprehensive description of amino acid metabolism and related transport proteins in different subtypes of breast cancer, providing a valuable reference for a deeper understanding of the metabolic mechanisms of breast cancer. However, there is still room for improvement in some areas, as detailed below.
- Specific problems and suggestions for improvement
(1) Completeness of literature references and reviews
Some research results are mentioned in part of the text, but the relevant literature is not clearly cited. For example, when describing the function of some amino acid transport proteins, there is a lack of corresponding literature support, which affects the credibility of the conclusions to a certain extent.
For some emerging or controversial ideas, there is insufficient comprehensive discussion with full citations from different research directions. For example, when discussing the relationship between certain transport proteins and breast cancer prognosis, only some supporting studies are cited, and no mention is made of the possible opposite views or the influence of other related factors.
Thank you for your advice. We have added links in the introduction. Highlighted in red.
(2) Research on innovation and depth
Although a relatively comprehensive summary has been made of the known roles of transmembrane amino acid transporters in breast cancer, it is lacking in terms of innovation. Most of the content is a consolidation of existing research results, and there is a lack of in-depth exploration of new discoveries or unique insights.
The potential mechanisms of some transport proteins in breast cancer metabolism are not explained in sufficient depth. For example, when discussing the interaction between transport proteins and signal pathways, only a superficial description of the association is provided, without exploring in depth the regulatory mechanism and dynamic process at the molecular level.
Our work consolidates the latest discoveries and achievements regarding the functional characteristics of all molecular biological subtypes of breast cancer. We discussed the latest studies regarding the expression of all amino acid transporter proteins that have already been studied and are potentially subject to study. We provided a complete list of genes that encode transporter proteins. We identified a complete list of miRNAs that affect the expression of amino acid transporter proteins. We also discussed the latest research on the relationship between immune regulation and amino acid metabolism. We considered possible targets for targeted therapy and provided examples of the latest research in this area. We compared a complete list of all cell lines that are currently known with molecular biological phenotypes of breast cancer. In conclusion, we outlined potential directions for future research. The novelty of this work is a new approach to considering amino acid metabolism in different breast cancer subtypes, considering miRNA expression and immune regulation. We offer a complex and most complete picture of the features of breast cancer metabolism. The uniqueness of this work lies in the systematization of comprehensive information on the metabolic and functional features of breast cancer. We identify new research paths that no one has mentioned before. For researchers, we provide a complete list of cell lines.
(3) Logical structure and clarity of expression
The overall logical structure of the article is relatively clear, but in some chapters, the organisation of the content is slightly loose. For example, when introducing the metabolic characteristics and transport proteins of different subtypes of breast cancer, there is some repetition of expressions and scattered information, which affects reading fluency.
The use of some technical terms and abbreviations is not fully explained, which may cause difficulties for readers who are not experts in the field. For example, the names of some signal pathways (such as PI3K/Akt/mTOR) and transporter families (such as SLC6) that frequently appear in the text should be fully explained when they first appear.
Thank you for your comment. We have corrected the structure. The missing terms have been deciphered and highlighted in red.
(4) Graph and chart quality and data analysis
The graphs and charts provided in the text (e.g., Figs. 2–5) help visually demonstrate the relationship between cell lines, transmembrane transport proteins, and amino acids. However, the quality and labeling of the graphs and charts could be improved. For example, the font in some graphs and charts is small, and the lines and markers are not clear enough, which affects the communication of information.
There is a lack of in-depth analysis and interpretation of the data in the figures. The correspondence between transport proteins and cell lines is simply listed, and the biological significance and potential laws behind the data are not further explored. Statistical methods can be used to analyse the data, such as comparing whether the differences in the expression levels of transport proteins in different subtypes are statistically significant, and the results of the analysis should be elaborated in the text.
Thank you for your advice. We have changed the appearance of the figures. The font has been increased for better perception. The content of the figures has not been changed. The original and main purpose of the figures was to simplify the perception of a large list of cell lines and transmembrane amino acid transporters for each molecular biological subtype of breast cancer.
Deep mechanisms of the relationship between amino acid metabolism and transporter proteins are discussed in Section 8 and in the discussion sections of each breast cancer subtype.
(5) Discussion of the clinical application value of the research
Although the paper mentions the significance of the research results for the development of treatment targets, there is a lack of specific discussion and prospects on how to translate these research findings into actual clinical applications. For example, there is no detailed explanation of how to design more effective treatment strategies for breast cancer based on the characteristics of transport proteins, as well as the challenges and solutions that may be faced in clinical practice.
We have considered your comment and added a new section 10 on prospective directions of targeted therapy.
(6) Conclusion section improvement
Although the conclusion section summarises the main content of the research, it lacks a high-level summary and sublimation of the research results. It fails to highlight the important contribution of the research in the field of breast cancer research and provide a clear guide for future research directions.
We have considered your comment and adjusted the conclusion (highlighted in red).
We have focused on the novelty of the work and new directions of practical research.
(7) Language and grammar issues
There are some places in the text where the language is not precise or fluent enough, and some sentences have complex structures, which affect the reader's understanding. For example, there are grammatical errors or unclear logic in some long sentences.
There are a few problems with inappropriate word choice, such as the use of some technical terms that are not precise enough or inappropriate in a particular context.
Thank you for your comment. We have made changes to the text.
(8) Discussion of the limitations of cell line research
The limitations of using cell lines as a research tool were not adequately discussed. Cell line research may not fully simulate the complex environment of tumours in vivo, which may affect the extrapolation of research results, but this was not mentioned in the article.
Thank you for your comment. We have created new section 11 research restrictions. The main restrictions include the lack of practical research in the field of studying amino acid transporters, their physical and biochemical characteristics. In addition, not all transport proteins have been studied in relation to oncological diseases, including breast cancer. We identified restrictions in the use of cell lines and the lack of extensive studies regarding a deep understanding of the mechanism of miRNA influence on the associated expression of transmembrane amino acid transporters in breast cancer. There are no studies in the field of the relationship between the metabolic landscape of amino acids in breast cancer cells for each individual subtype and immune cells included in the tumor microenvironment.
(10) Timeliness of data updates
Some of the cited literature is relatively old and may not reflect the latest research developments in the field. In the rapidly developing field of biomedicine, the timeliness of data is crucial to the quality of the review.
Thank you for your advice. We have replaced the links to old literary sources with new ones. Highlighted in red.
(11) Systematic research on the function of transmembrane transport proteins
There is a lack of systematic integration and analysis of the function of some transmembrane transport proteins. For example, when introducing different families of transport proteins, their role in different subtypes of breast cancer is described separately, but the synergistic or antagonistic effects of these transport proteins in the metabolic network of breast cancer are not discussed in depth.
Thank you for your comment. In this paper, we provide some examples of synergistic interactions between amino acid transporters.
In the section about luminal subtypes we wrote about SLC6A14 и SLC 1A2, lines 234-239: “The functioning of SLC6A14 and SLC1A2 transporters is interconnected. With increased expression of miR-23b-3p, inhibition of SLC6A14 activity occurs, but the activity of SLC1A2 significantly increases. As is known, SLC1A2 provides an influx of aspartate and glutamate, which support the proliferation of cancer cells. Such data were obtained when studying the metabolism of cancer cells in breast cancer patients with resistance to endocrine therapy”.
In the section about HER2+ lines 257-258: “Expression of SLC1A5 and SLC7A5 is coupled to, but not required for, function for individual transporter proteins”.
In the section about TNBC lines 280-281: “It is known that the activity of SLC 3A1 and SLC7A11 is coupled in the transport of cysteine”.
Lies 316-318: “The activity of SLC 1A5 and SLC7A11 is coupled. 7A11 is an antiporter. Through it, cystine enters the cell. Because of biochemical reactions, cysteine ​​is converted into cystine. Then, through the 1A5 transporter, cystine is exchanged for glutamate”.
Lines 320-321: “As it was mentioned before in HER2+ subtype of BC, the activity of SLC7A5 is coupled with SLC1A5. The deep mechanism of this coupled faction still has not been investigated”.
Lines 395-396: It is assumed that transporters 38A1 and 38A2 are functionally coupled. In-depth studies in this direction have not been conducted.
The description of the relationships in this review is incomplete since there is no research in this area.
Although the heterogeneity of breast cancer and the differences in the expression of transport proteins are mentioned in the article, there is a lack of in-depth discussion of the intrinsic connection and molecular mechanism between this heterogeneity and the expression of transport proteins. For example, it is not elaborated in detail how the differences in the genetic background and epigenetic regulation of breast cancer cells of different subtypes affect the expression and function of transport proteins.
In each section on individual molecular biological subtypes of breast cancer, we have provided a detailed discussion of the causes of expression of certain amino acid transporters. Discussion of immune regulation, genetics, and epigenetics is presented in separate sections. If our explanations seem incomplete, this is due to insufficient research in this area. Only the SCL1 and SLC7 transporters have been well studied. We provide the most complete list of all transporter proteins. We hope that our work will be a reason for more in-depth studies of the functions of all the listed amino acid transporter proteins.
(13) Expanding the relationship between immune regulation and amino acid metabolism
With the development of tumor immunotherapy, the role of immune regulation in the treatment of breast cancer has received increasing attention. However, the relationship between amino acid metabolism and immune regulation is not discussed in the paper. Amino acid metabolism not only affects the growth and proliferation of cancer cells, but may also affect the function of immune cells in the tumor microenvironment. However, this is not fully discussed in the paper.
Thank you for your advice. We have added a new section 8, which discusses the relationship between amino acid metabolism and immune regulation. In this section, we also outline potential ways for future research.
(14) Integration of the regulation of transport proteins by non-coding RNAs
Non-coding RNAs play an important role in the regulation of gene expression, but the paper does not systematically integrate the regulatory effects of non-coding RNAs (such as miRNAs and lncRNAs) on transmembrane amino acid transport proteins. Existing studies have shown that non-coding RNAs can regulate the expression of transport proteins through various mechanisms, affecting the metabolism of breast cancer cells, but these contents are not fully reflected in the paper.
Thank you for your advice. We added a section on genetics and epigenetics. We have provided a complete list of miRNAs that can affect the expression of transporter proteins. A full analysis of this topic is limited by the lack of sufficient practical research. In addition, the volume of the literature review does not allow a detailed description of the effect of miRNA on transporter proteins and their deep relationships. This may be the topic of future studies.
Round 2
Reviewer 2 Report
Comments and Suggestions for Authors
I don't have any doubts